# Characterization of the Aroma of an Instant White Tea Dried by Freeze Drying

**DOI:** 10.3390/molecules25163628

**Published:** 2020-08-10

**Authors:** Hui Ni, Qing-Xiang Jiang, Ting Zhang, Gao-Ling Huang, Li-Jun Li, Feng Chen

**Affiliations:** 1College of Food and Biological Engineering, Jimei University, Xiamen 361021, China; nihui@jmu.edu.cn (H.N.); 18370736681@163.com (Q.-X.J.); yczkting@126.com (T.Z.); hgaol@jmu.edu.cn (G.-L.H.); fchen@clemson.edu (F.C.); 2Key Laboratory of Food Microbiology and Enzyme Engineering Technology of Fujian Province, Xiamen 361021, China; 3Research Center of Food Biotechnology of Xiamen City, Xiamen 361021, China; 4Department of Food, Nutrition and Packaging Sciences, Clemson University, Clemson, SC 29634, USA

**Keywords:** instant white tea, GC-MS, GC-MS-O, aroma reconstruction and omission test, synergistic interactions

## Abstract

The aroma of an instant white tea (IWT) was extracted through simultaneous distillation–extraction (SDE) and analyzed by sensory evaluation, gas chromatography-mass spectrometry-olfactometry (GC-MS-O), aroma reconstruction, omission test and synergistic interaction analysis. Sensory evaluation showed the IWT was dominated with floral and sweet notes. The SDE extract had the aroma similar to the IWT. The main volatile components in the SDE extract were benzyl alcohol, linalool, hotrienol, geraniol, α-terpineol, coumarin, camphene, benzeneacetaldehyde, 2-hexanone, *cis*-jasmin lactone and phenylethyl alcohol. GC-MS-O and aroma reconstruction experiments showed 16 aroma-active compounds. Linalool, *trans*-β-damascenone and camphene were the major contributors to floral, sweet and green notes based on flavor dilution analysis and omission test. Linalool and *trans*-β-damascenone had synergistic effect to promote floral and sweet notes. Camphene and *trans*-β-damascenone had synergistic effect to reduce green and sweet notes. The study helps to understand the aroma of IWT and antagonism interactions among aroma-active volatiles.

## 1. Introduction

Various teas are popularly consumed around the world [1], due to its unique pleasant flavors and outstanding health benefits [2,3]. In addition, teas could be processed to instant teas via water extraction, concentration and spray/freeze drying. In decades, the market of instant teas keeps on increasing rapidly, because it is convenient to carry and easy to use in food industry.

The aroma is an essential characteristic that influences the perceived quality, price and consumer acceptance of tea and instant tea products [4]. In addition, the aromatic volatiles play important roles in regulation of human health [5,6]. Aromatic volatiles could be extracted by direct solvent extraction, Soxhlet extraction, simultaneous distillation–extraction (SDE), solvent-assisted flavor evaporation, headspace solid-phase microextraction, etc. [7]. Among them, SDE combines the advantages of liquid/liquid and steam distillation–extraction, and is convenient, simple for extracting high boiling-point volatiles in thermal processed products such as fried foods and instant teas [7,8]. Aroma characteristic could be evaluated by sensory evaluation [4,9,10,11] and instrument analysis such as GC-MS determination [8,9,12,13]. In addition, the aroma-active compounds could be determined with olfactometric techniques [14,15,16] and odor active value (OAV) analysis [10,11], as well as confirmation via aroma reconstruction and omission test [11,14,17]. Volatile compounds with concentrations over thresholds could contribute to tea aroma with their respective odors [18], as well as their antagonism and synergism interactions [12]. Moreover, manufacturing processes of tea and instant tea have noticeable effects on volatile compounds and overall aroma profiles of tea products [19]. In current, over 600 volatile compounds have been reported in tea and tea products [1], however, the understanding is still limited about the effects of antagonism and synergism interactions among volatiles on overall aroma.

White tea is a kind of slightly fermented tea. It is made from tea leaves only by a prolonged withering process and a drying process [9,20]. White tea has protective effects against cardiovascular diseases, cancer, diabetes mellitus, obesity, central nervous system and microorganism-based diseases [21]. In addition, white tea has unique sweet and umami tastes as well as smoothly fresh and green odors [20]. The market and production of white tea is increasing in recent years due to the desirable flavor and excellent health benefits. Its production was 5140 tons in China in 2009, while rapidly reached 15,700 tons in 2014 [21]. The main volatile compounds in white tea were hexanal, linalool, 2-methyl-butanal, phenylethyl alcohol, benzaldehyde, benzene acetaldehyde, (*Z*)-3-hexen-1-ol, 1-hexanol, (*E*)-2-hexenal, geraniol, etc. [20]. In recent years, more and more white teas are made into instant white tea (IWT) through hot water extraction, vacuum/reverse osmosis concentration, spray/freeze/vacuum drying [22]. Among IWTs, the IWT dried with freeze drying possess a more comfortable aroma in comparison to the IWT dried by spray drying. More than 50 kinds of flavor substances are determined in white teas from Fujian Province of China [23]. Although researchers have performed studies on the aroma characteristic of white tea, the aroma characteristics of IWT products made from white tea have been seldom elucidated.

On this context, this study aimed to: (1) investigate the aroma profiles and aromatic contributors of a IWT dried by freeze drying via SDE extraction, GC-MS and GC-O analysis, aroma reconstruction, omission test and synergistic interaction analysis; (2) investigate the effects of antagonism and synergism interactions among volatiles on overall aroma. This study elucidates contributions of volatiles and their interactions on the aroma of IWT, and thus helps in-depth understanding of the aroma of tea and tea products.

## 2. Results and Discussion

### 2.1. Sensory Evaluation

The IWT had the sensory scores of 5.5, 4.7, 3.7, 3.0, 2.2 and 2.3 for sweet, floral, green, roasted, woody and fruity notes (Figure 1), respectively. The SDE extract of IWT had the sensory scores of 4.6, 4.3, 3.2, 2.6, 2.7 and 2.0 for sweet, floral, green, roasted, woody and fruity notes (Figure 1), respectively. Both the SDE extract and IWT were dominated in floral, sweet and green notes, as well as weak roasted, woody and fruity notes. Obviously, the SDE extract had an overall aroma profile similar to that of IWT, indicating the characteristic aromatic volatiles of the IWT was successfully extracted by the SDE procedure. In previous, Qi et al. showed that fresh white tea was dominated with delicate and grassy green aroma, and a rapid aged white tea was characterized by its sweet aroma [9]. Perez-Burillo et al. indicated that a white tea brewed for 7 min was dominated by “floral”, “fruity” and “green” attributes [24]. The present SDE extract and IWT had the characteristic aroma different from those of fresh, slightly brewed whites, but the dominating sweet note seemed similar to that of the aged white tea as reported in previous literatures [9,24]. Although few literatures currently report the effects of instant tea preparation on the aroma of IWT, it has been shown that some factors such as heating and oxidation during instant tea processing could affect the sweet, green and floral odors of tea products [4,25]. For instance, when Pu-erh tea was processed into instant Pu-erh tea, the sweet attribute significantly increased [4]. In addition, when oolong tea was processed into instant oolong tea, the intensities of sweet, fruity and roasted notes increased significantly, but the intensities of floral and grass notes decreased significantly [25]. Therefore, the IWT and SDE extract had a dominated sweet note, which might be attributed to the heating and oxidation process during instant tea preparation.

### 2.2. GC-MS Analysis of Volatile Compounds in SDE Extract of IWT

It was reported that the main volatile compounds in the white tea were alcohols and aldehydes, including (*Z*)-3-hexen-1-ol (30.2 μg/g), linalool (49.4 μg/g), (*E*)-linalool oxide (furanoid) (58.6 μg/g), (*Z*)-linalool oxide (furanoid) (29.2 μg/g), benzyl alcohol (18.4 μg/g), 2-phenylethanol (31.8 μg/g), (*E*)-2-hexenal (24.6 μg/g), hexanal (34.7 μg/g) and benzeneacetaldehyde (19.8 μg/g) [20,26]. Among them, hexanal, (*E*)-2-hexenal and (*Z*)-3-hexen-1-ol were the characteristic compounds of a white tea [26]. Twenty-five volatile compounds were identified in the SDE according to the retention indices (RI) and mass spectrum by using Rtx-5ms and Rtx-wax columns. These volatiles included 10 alcohols, 2 aldehydes, 4 ketones, 5 oxides and 4 others. Among the 25 volatiles, 20 compounds were quantitatively analyzed by using their respective standard curves, and the relative content of other 5 compounds were estimated by the internal standard method due to lacking of standards (Table 1). The main volatile compounds of the SDE extract were benzyl alcohol (179.7 mg/L), linalool (128.7 mg/L), hotrienol (70.0 mg/L), geraniol (64.7 mg/L), α-terpineol (63.0 mg/L), coumarin (45.3 mg/L), camphene (42.1 mg/L), benzeneacetaldehyde (38.0 mg/L), 2-hexanone (31.4 mg/L), *cis*-jasmin lactone (24.6 mg/L), phenylethyl alcohol (19.4 mg/L), β-ionone epoxide (16.2 mg/L) and *cis*-linalool oxide (9.0 mg/L) (Table 1). By comparison, some characteristic aldehydes and alcohols in white teas, including hexanal, (*E*)-2-hexenal and (*Z*)-3-hexen-1-ol had not been detected in the SDE extract. This phenomenon could be attributed to aldehydes and alcohols being apt to evaporation and oxidation during preparation of instant tea as reported in a previous study [19]. In the future, more in-depth studies are needed to elucidate the transformation pathway.

In addition, the SDE extract seems to have higher content of geraniol, benzeneacetaldehyde and coumarin than the content of white teas reported by previous researches [20,26]. Geraniol and linalool are synthesized in tea leaves from the geranyl pyrophosphate (GPP) via the catalysis by terpene synthases (TPS) [18]. Benzeneacetaldehyde and coumarin are important derivatives of phenylalanine produced via the oxidation in tea leaves [18,27]. Tea infusions that are prepared with hot water from leaves have dramatically high contents of alcoholic volatiles such as geraniol [28], due to the thermal hydrolysis of glycosides precursors [18]. Heating during tea and instant tea processing may accelerate the oxidative degradation of phenylalanine [29], leading to higher contents of benzeneacetaldehyde and coumarin. Thus, the high content of geraniol, benzeneacetaldehyde and coumarin in the SDE extract could be attributed to the hydrolysis of glycosides precursors and oxidative degradation of phenylalanine caused by the heating during instant tea processing.

### 2.3. GC-MS-O and OAV Analyses of Aroma-Active Volatiles in the SDE Extract

Previously, the major aroma-active compounds of a white tea have been shown to be (*E*,*E*)-2,4-nonadienal, hexanal, (*E*,*Z*)-2,6-nonadienal, (*E*,*E*)-2,4-decadienal, decanal, benzeneacetaldehyde, 2-pentylfuran, linalool, dimethyl sulfide, β-ionone, etc., in view of the higher aroma character impact (ACI, a ratio of OAV in a mixture) factors [26]. These aroma-active compounds are mainly aldehydes with six to ten carbons having green, fatty or fresh notes [30]. In order to investigate the aroma-active volatiles of the IWT, the SDE extract was further analyzed using gas chromatography-mass spectrometry-olfactometry (GC-MS-O) combined with aroma extract dilution analysis (AEDA) and OAV determination. The SDE extract were sniffed to have 16 volatiles with noticeable notes based on GC-MS-O analysis (Table 2). These aroma-active compounds included 7 alcohols, 3 ketones, 2 aldehydes, 2 oxides, 1 hydrocarbon and 1 other. Among these volatiles, camphene (green note, FD = 64), linalool (floral note, FD = 64), 2-hexanone (floral and sweet notes, FD = 16), *trans*-β-damascenone (sweet note, FD = 64), benzeneacetaldehyde (sweet and honey notes, FD = 16) and safranal (green and woody notes, FD = 16) offered higher FD factor according to the AEDA analysis (Table 2). Meanwhile, most of these compounds had high OAVs, e.g., linalool, benzeneacetaldehyde, *trans*-β-damascenone and geraniol has OAVs of 21,451, 9505, 4,546,692 and 1616, respectively (Table 2). Despite some volatiles, e.g., hotrienol, α-terpineol and geraniol had low FDs showed high OAVs, which could be related to the different thresholds in water and air [11]. In short, the GC-MS-O and OAV analyses indicated that camphene, linalool, *trans*-β-damascenone, 2-hexanone, benzeneacetaldehyde, safranal, 3-hexanol, *cis*-linalool oxide, *trans*-linalool oxide, hotrienol, geraniol, benzyl alcohol, α-terpineol, phenylethyl alcohol, *trans*-β-ionone and indole were the major aroma contributors of the SDE extract. These aroma-active compounds were mainly alcohols as well as some aldehydes, which is obviously different from the aroma-active compounds of the white tea which with the aroma-active compounds mainly consisting of aldehydes with six to ten carbons [30].

### 2.4. Aroma Reconstruction and Omission Experiments

The 16 aroma-active compounds were identified via GC-MS-O analysis (Table 2). Only 15 of them (except hotrienol) were used to prepare the reconstruction model (FM1) according to their concentrations (Table 1). As shown in Figure 2A, FM1 and the SDE extract were detected to have similar aroma profiles, indicating the 15 volatiles are major aroma-active compounds of the SDE extract. In addition, FM1 and the SDE extract showed slight differences in aroma intensities in floral and roasted notes (*p* < 0.05), which might be ascribed to the lack of hotrienol. Meanwhile, there might be some compounds that could affect the aroma profile have not been identified yet, in view of researchers have shown that volatile compounds below the threshold concentrations can sometimes exert a remarkable influence on the overall aroma [32]. Recently, comprehensive two-dimensional gas chromatography time-of-flight mass spectrometry (GC × GC-TOFMS) has been shown to analyze aromatic contributors with advantages of high separation, fast data acquisition rate, wide linear dynamic range and full-range mass sensitivity [4]. Further, GC × GC-TOFMS coupled with sensory analysis are to be used to conduct further analysis.

The perception of an aroma compound is a complex process and affected by many factors, such as volatility, concentration, antagonism and synergism between aroma compounds [12,32]. For investigating how the volatiles affected the overall aroma profiles, three aroma active compounds, i.e., linalool (floral note), *trans*-β-damascenone (sweet note) and camphene (green note) were chosen to do omission test from FM1, because these compounds had the highest FD factor (FD = 64) among all the detected volatiles and possessed different notes. FM2, FM3 and FM4 were designed to omit the linalool, *trans*-β-damascenone and camphene from FM1, respectively. As shown in Figure 2B, in comparison to FM1, FM2 showed decreased intensities in the floral note (5.2 vs. 3.8) (*p* < 0.05) and sweet note (5.0 vs. 4.7), as well as the increased intensity of woody note (2.4 vs. 2.9). FM3 that was the model prepared by the omission of *trans*-β-damascenone from FM1, showed decreased intensities in sweet (5.0 vs. 3.8) (*p* < 0.05), floral (5.2 vs. 4.9) and fruity (2.5 vs. 2.1) notes, as well as slight increase in green note intensity (3.6 vs. 3.8). FM4 that was the model prepared by omission of camphene from FM1, showed decreases in the green (3.6 vs. 2.9) (*p* < 0.05) and roasted (1.7 vs. 1.4) notes, as well as increased intensity in floral note (5.2 vs. 5.4). These results indicated that linalool, *trans*-β-damascenone and camphene could affect the aroma not only through contributing the respective characteristic notes, but also via synergism interactions.

### 2.5. Interactions among Major Aroma Contributors

Interactions such as antagonism and synergism have been illustrated among aromatic compounds [12,32]. As a result that linalool, *trans*-β-damascenone and camphene had the highest FD factors (FD = 64) among all the detected volatiles possessing different notes, these three compounds were chosen to explore the interaction between different notes. In order to further explore how interactions among linalool, *trans*-β-damascenone and camphene affect the aroma profiles, omission tests were conducted using artificial solutions consisting of linalool, *trans*-β-damascenone and camphene with concentrations detected in the SDE extract. As shown in Figure 3A, the interaction model of linalool and *trans*-β-damascenone showed higher intensities in both floral (3.6 vs. 1.7) and sweet (3.0 vs. 1.9) notes in comparison to pure solutions containing only linalool or *trans*-β-damascenone (*p* < 0.05) (Figure 3A). The interaction model of linalool and camphene had similar floral note (1.7 vs. 1.2) in comparison to the pure solution of linalool, as well as similar green note (2.8 vs. 2.2) in comparison to the pure solution of camphene (Figure 3B). The interaction model of *trans*-β-damascenone and camphene had a decreased sweet note in comparison to the pure solution of *trans*-β-damascenone (1.2 vs. 1.9, *p* < 0.05), as well as weakened green note in comparison to the pure solution of camphene (1.1 vs. 2.8, *p* < 0.05) (Figure 3C). In addition, the interaction model of linalool, *trans*-β-damascenone and camphene had higher intensity of floral note than the interaction model of linalool and camphene (*p* < 0.05), as well as a higher intensity of sweet note than the interaction model of *trans*-β-damascenone and camphene (*p* < 0.05) (Figure 3D); Furthermore, the interaction model of linalool, *trans*-β-damascenone and camphene had the intensity of green note similar to the interaction model of *trans*-β-damascenone and camphene (Figure 3D); and the interaction model of linalool, *trans*-β-damascenone and camphene had a weaker green note than the interaction model of linalool and camphene (*p* < 0.05) (Figure 3D). These results demonstrated that linalool and *trans*-β-damascenone had positive synergism interactions to promote the floral and sweet notes; linalool and camphene had no noticeable interaction to affect the odor of either linalool (floral) or camphene (green); and *trans*-β-damascenone and camphene had an antagonism interaction to reduce odor intensities of both *trans*-β-damascenone (sweet) and camphene (green).

Previously, on one hand, Atanasova et al. found that sub- and peri-threshold concentrations of woody compounds modified the perception of a supra-threshold fruity odor [33]; Qin et al. illustrated that the fruit note from 6-methyl-5-hepten-2-one could promote the perception of sweet in tea infusions [34]. Compounds with similar structures or odors are apt to present synergistic/additive interactions to promote their odors [32]. For instance, 2-methylpropanal, 2-methylbutanal and 3-methylbutanal showed strong addition action in a grape wine [35]; hexanal, (*E*)-2-hexenal, (*E*,*E*)-2,4-heptadienal and (*E*)-2-heptenal presented a synergistic effect in the green note of oolong tea sample [32]. To our best knowledge, it is the first time to find that the synergism interaction of linalool and *trans*-β-damascenone positively promote both floral and sweet notes. On the other hand, Zhu et al. reported that the masking effects existed between (*E*)-2-hexenal (green note) and *β*-ionone (floral note) [32]. Perceptual suppression was common in mixtures with a large number of compounds as compared to binary mixtures [35]. Compounds with different structures frequently demonstrated masking effects [32]. For instance, (*E*)-2-hexenal (green and grassy note) could reduce the intensities of roasted and sulfur notes in the oolong tea [32]; Zhang et al. showed that *trans*-β-ionone can significantly eliminate stale note and improved the overall aromatic acceptance of instant ripened Pu-erh tea infusion [36]. By comparison, it is the first time to reveal the synergistic interaction of *trans*-β-damascenone and camphene could reciprocally reduce the notes of both *trans*-β-damascenone (sweet) and camphene (green).

In short, we found linalool and *trans*-β-damascenone had a synergism interaction to promote both floral and sweet notes; and *trans*-β-damascenone and camphene had an antagonism interaction to reduce sweet and green notes. Human olfactory receptors on the cilia of olfactory neurons have been proposed to explain and design the aroma characteristic by encoding odorant mixtures [37]. The interaction between aromatic compounds could be studied by stimulating the response characteristics of olfactory receptor meridians and monitoring changes of cell calcium ions and electrophysiology [37,38]. In the future, interactions among aromatic compounds studies might be focused on how the interactions affect olfactory receptors binds with aroma-active chemicals.

## 3. Materials and Methods

### 3.1. Materials

A typical IWT was prepared with freeze drying from the mixture of five batch of fresh white teas in Fujian Da Ming Development Company (Zhangzhou, Fujian province, China). White tea was smashed to filtrate through an 80 mesh sieve. The resultant white tea powder was extracted using 18-folds of water (V/W) at 95 °C for 30 min. After cooling to 30–40 °C, the extract was concentrated to the content of 10–12 Brix using ultra filtration and reverse osmosis at 40 °C, then freeze-drying to powder to get the IWT sample.

### 3.2. Chemicals

Standard 2-hexanone, 1*H*-1-ethyl-pyrrole, benzeneacetaldehyde, *cis*-linalool oxide, *trans*-linalool oxide, linalool, 3-octen-2-ol, phenylethyl alcohol, α-terpineol, safranal, camphene, geraniol, indole, 3-hexanol, hexanal, 1-octanol and 2,5-dimethylpyrazine were purchased from Sigma Co. Ltd. (St. Louis, MO, USA). Benzyl alcohol, *trans*-β-damascenone, *trans*-β-ionone, 2, 4-ditert-butylphenol, cedrol and caryophyllene oxide were purchased from Alfa Aesar Co. Ltd. (Heysham, Lancashire, UK). Standard chemical series of C_8_-C_20_ alkanes that were used to determine the liner retention index (RI) and the internal standard (Cyclohexanone) were obtained from Sigma Co. Ltd. (St. Louis, MO, USA). The other chemicals were purchased from Sinopharm Chemical Reagent Co. Ltd. (Shanghai, China).

### 3.3. Preparation of the SDE Extract of IWT

Volatile compounds in IWT were extracted according to the methods reported in the literature [36]. An SDE apparatus similar to the design of Lickens–Nickerson apparatus was purchased from Beijing Glass Instrument Factory (Beijing, China). Thirty grams of the IWT sample was immersed in a 500 mL flask with 300 mL of distilled water, and 100 mL of hexane applied as extraction solvent was placed in another flask. Both flasks were placed in the Lickens–Nickerson apparatus. After being respectively heated up to the boiling points of water and hexane, the extraction with reflux was continued for 1.5 h to allow the volatiles to be collected in the organic phase. After being cooled to ambient temperature, the extract was collected and dried over anhydrous sodium sulfate overnight. The extract was concentrated approximately to the volume of 0.5 mL at room temperature by using a gentle stream of high-purity nitrogen, and adjusted to the volume of 1.5 mL with hexane. The concentrated extraction was stored at −20 °C temporarily before analysis with sensory evaluation, GC-MS and GC-O.

### 3.4. Sensory Evaluation

Before the quantitative descriptive analysis, panelists had discussed aroma compositions of samples through three preliminary sessions (each spent 3 h) until all of them agreed to the attributes, according to the statements from ISO 8589 and previous researches [25,39] with slight modifications. The green, floral, sweet, woody, roasted and fruity notes were selected as indicators for sensory evaluation. Samples were evaluated by fifteen panelists (six males and nine females), with ages between 20 and 30 years old. All the panelists were trained to distinguish the aroma characteristics and intensities using a series of standards solutions. In detail, standard solutions of hexanal (green) [25], linalool (floral) [25], benzeneacetaldehyde (sweet) [32], *trans*-β-Ionone (woody) [11], 2,5-dimethylpyrazine (roasted) [30] and 1-octanol (fruity) [11] were used to train panelists to get familiar with the aroma characteristic and intensity.

Two samples (SDE extract and white infusion) were sensory evaluated. An aliquot of 150 μL of the SDE extract was diluted with 150 μL of ethanol and 9.7 mL boiling water, and 0.1 g of IWT was dissolved in 10 mL boiling water for 5 min. The two samples were placed in an 80 °C water bath for 5 minutes, followed by the sensory evaluation in a clean environment under illumination at 25 ± 2 °C using a 9-point scoring method, in which 0 indicated an unperceived attribute intensity and 9 indicated a very strong attribute intensity. Each panelist separately gave a score within 0 to 9 for green, floral, sweet, woody, roasted and fruity notes. Each sample was evaluated three times by each panelist. Means of each sample were calculated.

### 3.5. GC-MS Analysis

Ten microliters of the internal standard cyclohexanone was added into 990 µL SDE extract. After that, 1 µL of the aliquot was injected for the gas chromatography couple with mass spectrum (GC-MS) analysis. A QP2010 GC-MS (Shimadzu, Kyoto, Japan) and two different fused silica capillary columns, i.e., Rtx-5MS (60 m × 0.32 mm × 0.25 µm, Restek Corporation, Bellefonte, PA, USA) and Rtx-Wax (60 m × 0.32 mm × 0.25 µm, Restek Corporation, Bellefonte, PA, USA) columns were used. The carrier gas was the 99.999% high purity helium. The column flow rate was 3 mL/min in the mode of splitless injection. The inlet temperature was 230 °C. The oven temperature was programmed from 50 °C for 2 min, then increased at a rate of 3 °C/min to 200 °C and held for 1 min. The temperatures of the ion source and the interface were 220 °C and 250 °C, respectively. The mass scan range of *m*/*z* was set from 35 to 500 amu.

Most of the volatiles were identified by matching their detected MS spectra and retention indices (RI) to those of standards on both columns, and were quantitatively analyzed according to their respective calibration curves on Rtx-5MS column using selective ion monitoring (SIM) mode. Yet some chemicals that lacked standards were tentatively identified based on matching ion fragment and RI values to those from MS Spectra Library (FFNSC1.3, NIST08, NIST08s) and reported in relevant references, and relative quantitative analysis through the internal standard method (with cyclohexanone as internal standard).

### 3.6. GC-MS-O Analysis

An Agilent 5975C-7890A GC-MS (Agilent Technologies, Palo Alto, CA, USA) was equipped with an olfactory detection port Gerstel ODP-2 (Gerstel AG Enterprise, Mülheim an der Ruhr, Germany). The GC was fitted with HP-INNOWAX column (60 m × 0.25 mm × 0.25 μm, Agilent, Palo Alto, CA, USA). 0.5 µL of the sample was injected into the GC-MS-O system in a splitless mode. The oven temperature was started at 40 °C for 1 min, and increased in a rate of 5 °C/min to 230 °C and then kept for 3 min. High purity nitrogen was used as the carrier gas at 1.8 mL/min. Temperature of the injector port was 250 °C.

GC-MS-O experiment was performed by three panelists (two females and one male). All the panelists were trained for 30 h over a period of 3 weeks using a series of standards solutions, so as to recognize, describe and discriminate the odors of different compounds. Each volatile was recorded with the retention time (RT), sniffed aroma attribute and the aroma intensity (AI). The AI was ranked in five levels, where “1” means extremely weak, “3” impresses medium, “5” means extremely strong [40]. The odorants were detected by at least two panelists were recorded. The experiment was replicated in triplicate by each panelist, and the AI was averaged.

### 3.7. Aroma Extract Dilution Analysis (AEDA)

AEDA was conducted based on the aforementioned GC-MS-O method with procedures reported in previous literatures [14,15,16]. The SDE extract were gradually diluted in the ratio of 4 (by volume) with n-hexane, followed by GC-MS-O analysis. The flavor dilution (FD) factor of each compound was determined as the maximum dilution number (recognized at least by two panelists) at which the odorant could be perceived. The experiment was replicated triplicate by each panelist.

### 3.8. Odor Activity Values (OAV) Analysis

The OAVs were calculated by dividing the calculated concentrations with sensory thresholds in water, which were reported in the literatures.

### 3.9. Aroma Reconstruction and Omission Test of Major Aroma Contributors

In order to evaluate whether the key aromatic compounds were identified correctly, aroma reconstruction and omission experiments were performed according to the methods reported in the previous literatures with minor modifications [11,14,17]. The aroma reconstruction model of the SDE extract (FM1) was prepared with the 15 aromatic volatiles (except hotrienol) at concentrations measured (Table 1), where n-hexane was used as the matrix, and the final volume was 1.5 mL. In order to further determine how the volatiles affect the overall aroma profile, three omission models (FM2, FM3 and FM4) were constructed by omitting the major volatiles that had FD values of 64 from FM1. FM2 was the model by omitting 128.7 mg/L of linalool from FM1. FM3 was the model by omitting 5.9 mg/L of *trans*-β-damascenone from FM1. FM4 was model by omitting 42.1 mg/L of camphene from FM1.

For evaluating aroma profiles of the aroma reconstruction and omission models, the aliquot of 150 μL of the FM1, FM2, FM3 and FM4 were diluted with 150 μL of ethanol and 9.7 mL boiling water, respectively. The diluted samples were heated in an 80 °C water bath for 5 minutes, followed by sensory evaluation using the aforementioned method of Section 3.4, each panelist separately gave a score within 0 to 9 for green, floral, sweet, woody, roasted and fruity notes. Each sample was evaluated three times by each panelist. Means of each sample were calculated.

### 3.10. Investigation of Interactions of Major Aroma Contributors

Because linalool, *trans*-β-damascenone and camphene had the highest FD factors (FD = 64) among all the detected volatiles possessing different notes, these three compounds were chosen to explore the interaction between different notes. To study the interactions among the major aroma contributors, seven interaction models were prepared with linalool, *trans*-β-damascenone, camphene and n-hexane to a final volume of 1.5 mL, according to a previous method [36]. The samples were: (1) the pure floral note model was the pure solution of linalool (128.7 mg/L); (2) the pure sweet note model was the pure solution of *trans*-β-damascenone (5.9 mg/L); (3) the pure green note model was the pure solution of camphene (42.1 mg/L); (4) the interaction model of floral and sweet notes was the mixed solution of linalool (128.7 mg/L) and *trans*-β-damascenone (5.9 mg/L); (5) the interaction model of floral and green notes was made of linalool (128.7 mg/L) and camphene (42.1 mg/L); (6) the interaction model of sweet and green notes was made of *trans*-β-damascenone (5.9 mg/L) and camphene (42.1 mg/L); and (7) the interaction model of floral, sweet and green notes was made of linalool (128.7 mg/L), *trans*-β-damascenone (5.9 mg/L) and camphene (42.1 mg/L). The concentrations in the models were identical to those detected in the SDE extract. All samples were sensory evaluated in terms of the aroma intensities by using the method described in Section 3.4, each panelist separately gave a score within 0 to 9 for green, floral and sweet notes. Each sample was evaluated three times by each panelist. Means of each sample were calculated.

### 3.11. Statistical Analysis

All of the experiments were conducted in triplicate. The calculations of average and standard deviation, and drawing radar chart and bar chart were performed by the Excel 2010 software. The significant analysis was performed by the software SPSS-IBM 19.0 software as well as Microsoft Excel 2010.

## 4. Conclusions

The freeze drying instant white tea and its SDE extract were dominated with floral and sweet notes, which is different from the aroma profiles of white tea leaves infusion. The SDE extract had volatile compounds different from white teas due to evaporation and oxidation of characteristic aldehydes and alcohols, as well as the hydrolysis of glycosidic precursors and oxidative degradation of phenylalanine. The aroma profile of SDE extract was attributed to the odors and interaction of 16 aroma-active compounds, e.g., camphene, linalool, 2-hexanone, *trans*-β-damascenone, benzeneacetaldehyde, safranal, etc. Linalool and *trans*-β-damascenone had a synergism interaction to promote both floral and sweet notes; and *trans*-β-damascenone and camphene had an antagonism interaction to reduce sweet and green notes. This study helps in-depth understanding of the aroma of tea and tea products. In the future, GC×GC-TOFMS coupled with sensory analysis are to be used to analyze volatiles existing at low concentrations; and human olfactory receptors on the cilia of olfactory neurons are to be applied to simulate the interactions among characteristic odorants.

## Figures and Tables

**Figure 1 molecules-25-03628-f001:**
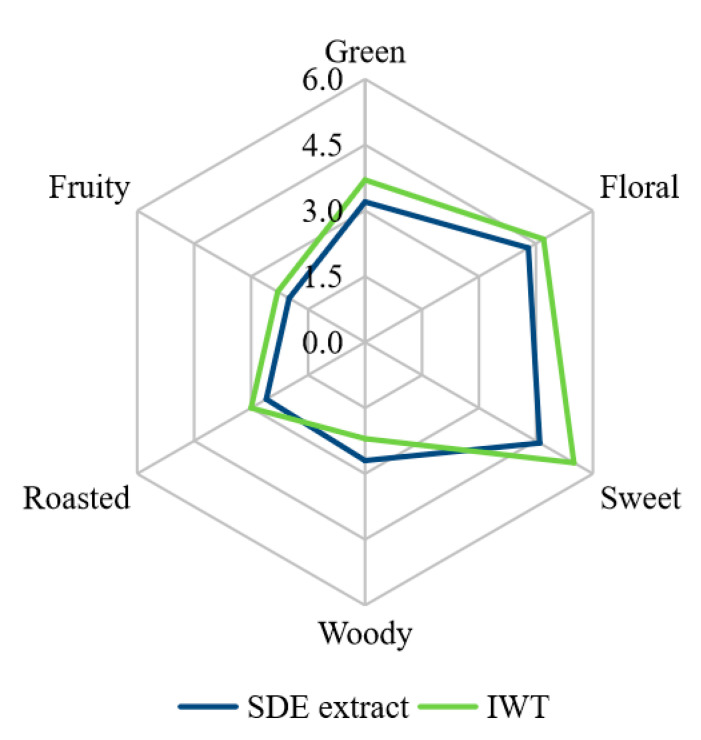
Sensory evaluation of simultaneous distillation–extraction (SDE) extracts and instant white tea (IWT) samples. SDE extracts and IWT samples showed significant difference (*p* < 0.05) in the sweet note.

**Figure 2 molecules-25-03628-f002:**
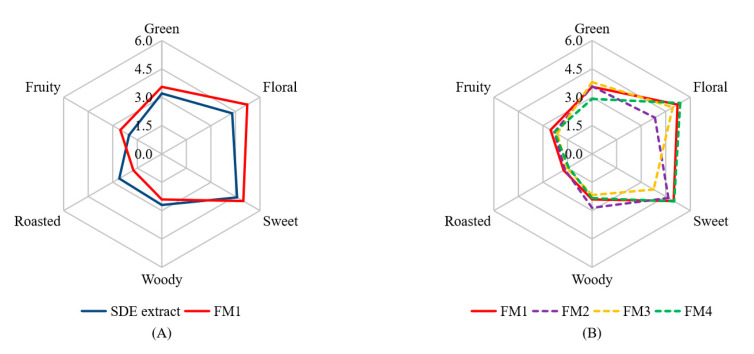
Aroma reconstruction (**A**) and aroma omission (**B**) based on GC-MS-O and OAV analyses of the SDE extract. FM1 was the aroma reconstruction model; FM2 was prepared by omitting 128.7 mg/L of linalool (floral) from FM1; FM3 was prepared by omitting 5.9 mg/L of *trans*-β-damascenone (sweet) from FM1; FM4 was prepared by omitting 42.1 mg/L of camphene (green) from FM1. In Figure 2A, SDE extracts and FM1 samples showed significant difference (*p* < 0.05) in floral and roasted notes; in Figure 2B, FM1 and FM2 samples showed significant difference (*p* < 0.05) in floral note, FM1 and FM3 samples showed significant difference (*p* < 0.05) in sweet note, FM1 and FM4 samples showed significant difference (*p* < 0.05) in green note.

**Figure 3 molecules-25-03628-f003:**
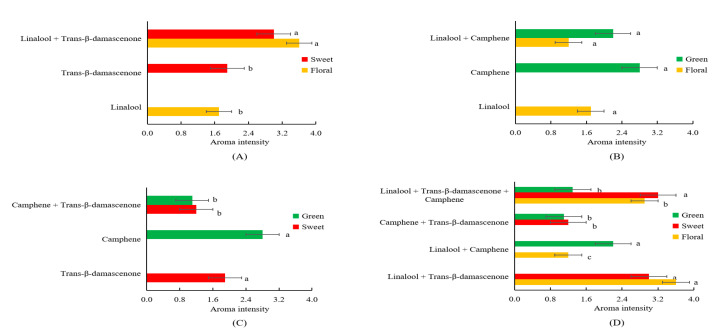
Interactions of linalool and *trans*-β-damascenone (**A**), linalool and camphene (**B**), *trans*-β-damascenone and camphene (**C**) and linalool, *trans*-β-damascenone and camphene (**D**). In each figure, different letters (a, b, c) behind columns in the same color represent significant difference in the aroma intensity (*p* < 0.05).

**Table 1 molecules-25-03628-t001:** Qualitative and quantitative analysis of volatile compounds in the SDE extract.

No	Volatile	RTX-5MS		RTX-WAX		Characteristic Ion Fragment	Reference *^e^*	Standard Curve *^f^*	*R* ^2^	Range (mg/L)	CF *^g^*	Concentration (mg/L) *^h^*
		RI *^a^*	RI *^b^*	RI *^c^*	RI *^d^*							
	alcohols											
1	3-hexanol	-	797	-	1211	56 69 84	Std MS	*Y* = 1.1043*X* − 0.0868	0.9906	0.1–10	0.732	2.7 ± 0.6
2	benzyl alcohol	1036	1037	1874	1877	108 79 107	Std MS R	*Y* = 0.1963*X* − 0.4311	0.9970	0.5–200	0.541	179.7 ± 9.7
3	linalool	1100	1100	1552	1552	71 41 93	Std MS R	*Y* = 2.2918*X* − 0.1747	0.9996	0.5–200	0.441	128.7 ± 3.4
4	Hotrienol ^i^	1105	1107	-	1623	71 83 43	MS R	internal standard method				70.0 ± 3.5
5	3-octen-2-ol	1108	-	1590	-	81 39 110	Std MS R	*Y* = 3.6322*X* − 0.2084	0.9991	0.5–100	0.277	3.3 ± 0.0
6	phenylethyl alcohol	1114	1114	1907	1912	91 92 122	Std MS R	*Y* = 7.6178*X* − 0.5345	0.9995	0.5–100	0.132	19.4 ± 1.1
7	*α*-terpineol	1192	1190	1695	1692	59 93 121	Std MS R	*Y* = 1.2469*X* − 0.0785	0.9994	0.5–100	0.808	63.0 ± 2.1
8	geraniol	1259	1260	1854	1854	69 41 68	Std MS R	*Y* = 1.6663*X* − 0.1813	0.9994	0.5–100	0.603	64.7 ± 4.5
9	2,4-ditert-butylphenol	1516	1513	-	2321	191 57 206	Std MS R	*Y* = 17.5469*X* + 0.1502	0.9998	0.1–10	0.057	0.2 ± 0.1
10	cedrol	1608	1608	-	2112	95 150 151	Std MS R	*Y* = 1.1834*X* − 0.0940	0.9995	0.5–100	0.867	1.0 ± 0.0
	aldehydes											
1	benzeneacetaldehyde	1044	1044	1628	1626	91 92 120	Std MS R	*Y* = 1.3974*X* − 0.0543	0.9995	0.5–100	0.719	38.0 ± 2.2
2	safranal	1201	1201	1631	1648	107 91 121	Std MS R	*Y* = 2.6516*X* − 0.0499	0.9998	0.5–100	0.378	5.0 ± 0.2
	ketones											
1	2-hexanone	-	792	-	1098	43 58 57	Std MS	*Y* = 0.2404*X* − 0.2621	0.9990	0.5–100	0.264	31.4 ± 2.9
2	*trans*-β-damascenone	1387	1386	1810	1810	177 69 41	Std MS R	*Y* = 1.3702*X* − 0.0799	0.9998	0.5–100	0.735	5.9 ± 0.2
3	*trans*-β-ionone	1490	1490	1930	1926	177 43 41	Std MS R	*Y* = 7.7843*X* − 0.1273	0.9995	0.1–10	0.129	0.2 ± 0.0
4	*cis*-jasmin lactone ^i^	1498	1497	2207	2226	99 71 55	MS R	internal standard method				24.6 ± 0.6
	oxides											
1	*cis*-linalool oxide	1072	1073	1439	1438	59 43 94	Std MS R	*Y* = 2.8420*X* − 0.0683	0.9997	0.5–100	0.354	9.0 ± 0.3
2	*trans*-linalool oxide	1088	1088	1468	1471	59 94 43	Std MS R	*Y* = 4.3092*X* − 0.0727	0.9997	0.5–100	0.233	7.0 ±0.2
3	*cis-pyranoid*-linalool oxide ^i^	1170	1167	1738	1742	68 94 43	MS R	internal standard method				7.4 ± 0.2
4	β-ionone epoxide ^i^	1491	1488	1982	2002	123 43 135	MS R	internal standard method				16.2 ± 0.6
5	caryophyllene oxide	1615	1613	-	2014	43 41 79	Std MS R	*Y* = 3.7303*X* − 0.3035	0.9995	0.5–100	0.271	3.2 ± 0.1
	others											
1	1*H*-1-ethyl-pyrrole	815	812	-	-	80 95 67	Std MS R	*Y* = 1.0108*X* − 0.0079	0.9996	0.5–100	0.990	3.8 ± 0.4
2	Coumarin ^i^	1225	1224	-	-	118 146 90	MS R	internal standard method				45.3 ± 2.3
3	camphene	1230	-	-	1059	93 121 79	Std MS R	*Y* = 0.2809*X* + 0.0015	0.9999	0.5–100	3.550	42.1 ± 3.1
4	indole	1299	1295	2345	2414	117 90 89	Std MS R	*Y* = 4.5914*X* − 0.4733	0.9994	0.5–100	0.221	6.5 ± 0.3

*^a^* RI is obtained by GC-MS analysis using the Rtx-5MS column. *^b^* RI is reported in the website (http://webbook.nist.gov/chemistry/) and is analyzed using a column similar to Rtx-5MS. *^c^* RI is obtained by GC-MS analysis using the Rtx-wax column. *^d^* RI is reported in the website (http://webbook.nist.gov/chemistry/) using a column similar to Rtx-wax. *^e^* Std indicates that the identification was confirmed by matching a standard, and R is referred to the database on the web (http://webbook.nist.gov/chemistry/). *^f^* All of the equations of the calibration curves of authentic standard chemicals (ASCs) are calculated in the SIM mode, where X is the ratio of the concentration of the ASC to that of the internal standard (IS) and Y is the ratio of the peak area of the ASC to that of the IS. *^g^* CF represents correction factors using this formula: CF = (As/Ms)/(Ar/Mr), As represents the corresponding quantitative ion (SIM mode) area of the IS, Ar is the corresponding quantitative ion (SIM mode) area of the ASC, Ms is the concentration of IS, Mr represents the concentration of the ASC. *^h^* The concentration mg/L represents how many micrograms of the volatile compound per liter of SDE extract. ^i^ The relative content of hotrienol, *cis*-jasmin lactone, *cis*-*pyranoid*-linalool oxide, β-ionone epoxide and coumarin were estimated by the internal standard method (with cyclohexanone as internal standard), due to the lack of standards.

**Table 2 molecules-25-03628-t002:** Gas chromatography-mass spectrometry-olfactometry (GC-MS-O) and odor active value (OAV) analyses of aroma-active compounds in the SDE extract.

No	RI	Volatiles	Aroma Description *^a^*	Aroma Intensity *^b^*	FD Factor *^c^*	Threshold (μg/L) *^d^*	OAV	Aroma Description *^e^*
1	1031	3-hexanol	roasted	3.7	4	NF *^f^*	-	fruity, alcoholic
2	1072	2-hexaone	floral, sweet	4.7	16	NF *^f^*	-	fruity
3	1075	camphene	green	4.7	64	450 [31]	94	green, camphoreous
4	1442	*cis*-linalool oxide	fruity	3.0	4	320 [30]	28	floral, woody
5	1460	*trans*-linalool oxide	fruity, floral	1.7	4	320 [10]	22	floral
6	1535	linalool	floral	5.0	64	6 [10]	21451	citrus, floral, sweet
7	1599	hotrienol	green, woody	2.3	4	110 [10]	637 *^g^*	floral, green, woody
8	1655	benzeneacetaldehyde	sweet, honey	4.7	16	4 [10]	9505	sweet, floral, honey-like
9	1687	safranal	green, woody	3.7	16	NF *^f^*	-	woody
10	1735	α-terpineol	green	1.3	1	330 [11]	191	floral
11	1832	*trans-*β-damascenone	sweet	5.0	64	0.0013 [30]	4546692	honey, sweet
12	1859	geraniol	sweet, floral	2.3	4	40 [11]	1616	green, floral
13	1876	benzyl alcohol	floral	2.0	4	10000 [10]	18	floral, rose-like
14	1909	phenylethyl alcohol	floral	1.2	1	1000 [10]	19	floral, rose-like
15	1930	*trans*-β-ionone	woody	1.4	1	7 [11]	29	floral, woody
16	2350	indole	floral	0.9	1	500 [11]	13	animal-like

*^a^* Aroma description from panelists. *^b^* Aroma intensity from panelists. *^c^* FD factor from panelists. *^d^* Odor thresholds in water. *^e^* The aroma description is derived from the literature corresponding to the threshold of the compound, and the aroma description of the compound whose threshold is not found is from the website (http://www.thegoodscentscompany.com/). *^f^* NF means the threshold values have not been found in references. *^g^* The content of the hotrienol was determined by internal standard method temporarily, and its OAV value might not be accurate, so the compound was removed during aroma reconstruction and aroma omission.

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
