# Peer review of "Characterization of the Aroma of an Instant White Tea Dried by Freeze Drying"

_molecules, 2020, doi:10.3390/molecules25163628_

Round 1

Reviewer 1 Report

Dear Authors;

The publication submitted for review entitled: “Characterization of the aroma of an instant white tea dried by freeze drying” by Hui Ni et al. is related to the study of the aroma profiles and aromatic contributors of instant white tea (IWT) dried by freeze drying by SDE extraction and quantitative analysis using GC-MS and qualitative analysis using GC-O. The obtained results were to enable evaluation of the effects of antagonism and synergism interactions among volatiles on overall aroma. The results and their interpretation bring new information to existing research. It should be noted that the work is very similar to another article by the Authors, which also appeared in the Molecules journal in 2019 (Molecules. 2019 Dec; 24 (24): 4473; doi: 10.3390 / molecules24244473), in which the material for research was instant ripened Pu-erh tea. From the point of view of scientific novelty, Authors should note what is innovative in their publication.

In order to improve the quality of the manuscript, Authors should explain:

  • Text (page 4, line 110-114) and Table 1 - If the linearity range is expressed in mg/L, why is the concentration given in μg/L. It is not understandable, especially as the concentrations are very high. Express concentration in mg/L. Similarly, throughout the text.
  • Table 1 should contain the value of the limit of detection (LOD) and quantification (LOQ), which is very important in determining linearity.
  • Table 1 - On what basis are the concentrations for hotrienol, cis-jasmin lactone, coumarin, cis-pyranoid-linalool oxide and β-ionone epoxide? The range of linearity was not determined for these relationships.
  • Page 12; line 178-182 – Please explain. The authors write that a model was constructed using 15 compounds without hotrienol (i.e. 14 compounds). Next sentence confirms the identification of 15 compounds.
  • Page 13 - Chapter 2.5 and 3.10 - On what basis were linalool, trans-β-damascenone and camphene selected as the main ingredients of the aroma?
  • What does "minor modifications [35]" of the method mean?
  • Page 15, line 274 - Why was 95oC used to extract the volatile compounds from the tea? From the consumer's point of view, the recommended brewing temperature is between 70 and 90 o
  • Page 15, line 296 – What was the temperature during the evaporation of hexane to a volume of 0.5 mL?
  • Page 15, line 297-298 - Is compared to loss of volatile compounds in the extract hexane to store the samples at -20 oC?

Author Response

Dear Editors and Reviewers:

Thank you for your letter and for the reviewer’s comments concerning our manuscript entitled “Characterization of the aroma of an instant white tea dried by freeze drying” (ID: molecules-872960). We have made careful modifications according to the comments. The revisions are marked in red color. The modifications and responds to the comments are stated below one by one:

Responds to the comments of Reviewer #1

General comment: The publication submitted for review entitled: “Characterization of the aroma of an instant white tea dried by freeze drying” by Hui Ni et al. is related to the study of the aroma profiles and aromatic contributors of instant white tea (IWT) dried by freeze drying by SDE extraction and quantitative analysis using GC-MS and qualitative analysis using GC-O. The obtained results were to enable evaluation of the effects of antagonism and synergism interactions among volatiles on overall aroma. The results and their interpretation bring new information to existing research. It should be noted that the work is very similar to another article by the Authors, which also appeared in the Molecules journal in 2019 (Molecules. 2019 Dec; 24 (24): 4473; doi: 10.3390 / molecules24244473), in which the material for research was instant ripened Pu-erh tea. From the point of view of scientific novelty, Authors should note what is innovative in their publication.

Response: Thank you very much for the valuable comment. The two studies are totally different. As recited in line 381-382 in the revision, Zhang et al. showed trans-β-ionone significantly eliminated stale note from methoxybenzenes and improved the overall aromatic acceptance of instant ripened Pu-erh tea infusion. The authors of present study found linalool and trans-β-damascenone had synergistic effect to promote floral and sweet notes, and camphene and trans-β-damascenone had synergistic effect to reduce green and sweet notes. In addition, experimental materials used in the two manuscripts are different.

  1. Comment: Text (page 4, line 110-114) and Table 1 - If the linearity range is expressed in mg/L, why is the concentration given in μg/L. It is not understandable, especially as the concentrations are very high. Express concentration in mg/L. Similarly, throughout the text.

Response: We appreciate you for the precious comment. We apologize for the inconsistent concentration units between the linear range and the standard curve. By following your comment, the concentration units have been revised to mg/L throughout the manuscript. The revision in page 4, line 111-115; Table 1; page 7, line 130; Figure 2, line 212-214; page 16, line 379-380 and line 392-399 were marked in red.

  1. Comment: Table 1 should contain the value of the limit of detection (LOD) and quantification (LOQ), which is very important in determining linearity.

Response: Thank you for your instruction. The value of the limit of detection (LOD) and quantification (LOQ) are indeed very important. The LOD and LOQ were determined in our previous study. The LOD and LOQ were not shown in this manuscript in order to save the space, because all the detected concentrations were in the range of linear regressive plots.

  1. Comment: Table 1 - On what basis are the concentrations for hotrienol, cis-jasmin lactone, cis-pyranoid-linalool oxide, β-ionone epoxide and coumarin? The range of linearity was not determined for these relationships.

Response: Thank you so much for your careful review. Due to the lack of standards for hotrienol, cis-jasmin lactone, cis-pyranoid-linalool oxide, β-ionone epoxide and coumarin, the concentrations of these compounds were temporarily estimated by the internal standard method (with cyclohexanone as internal standard). We apologize for the vague expression which has been noted using footnotes following Table 1 ‘iThe relative content of hotrienol, cis-jasmin lactone, cis-pyranoid-linalool oxide, β-ionone epoxide and coumarin were estimated by the internal standard method (with cyclohexanone as internal standard), due to the lack of standards’ (page 7, line 131-132).

  1. Comment: Page 12; line 178-182 – Please explain. The authors write that a model was constructed using 15 compounds without hotrienol (i.e. 14 compounds). Next sentence confirms the identification of 15 compounds.

Response: Thank you very much for pointing out the vague expression. Sixteen aroma-active compounds were identified by GC-MS-O. Only 15 of them were used to construct aroma recombination model, due to the lack of hotrienol standard.

In order to avoid misunderstanding, “The aroma-active compounds identified via GC-MS-O analysis (Table 2) were used to prepare the aromatic model according to their concentrations (Table 1). The resultant reconstruction model (FM1) was prepared with 15 aroma-active compounds as detected in the SDE extract, whereas hotrienol didn't included in FM1, due to lack of the standard” were changed to "The 16 aroma-active compounds were identified via GC-MS-O analysis (Table 2). Only 15 of them (except hotrienol) were used to prepare the reconstruction model (FM1) according to their concentrations (Table 1) (page 11, line 182-184).

  1. Comment: Page 13 - Chapter 2.5 and 3.10 - On what basis were linalool, trans-β-damascenone and camphene selected as the main ingredients of the aroma?

Response: Thank you very much for your comment. Linalool, trans-β-damarenone and camphor were selected as the main ingredients of the aroma according to these three compounds have the highest flavor dilution factor (FD = 64) among all the aroma-active volatiles. By following your comment, we added “Because linalool, trans-β-damascenone and camphene had the highest FD factors (FD = 64) among all the detected volatiles possessing different notes, these three compounds were chosen to explore the interaction between different notes” in chapter 2.5 and 3.10 (page 12, line 222-224; page 16, line 387-389).

  1. Comment: What does "minor modifications [35]" of the method mean?

Response: Thank you so much for your comment. By following your comment, the “To study the interactions among the major aroma contributors, seven interaction models were prepared with linalool, trans-β-damascenone and camphene according to a previous method with minor modifications [35]” was changed to “To study the interactions among the major aroma contributors, seven interaction models were prepared with linalool, trans-β-damascenone, camphene and n-hexane to a final volume of 1.5 mL, according to a previous method [35]” (page 16, line 389-391). The "minor modifications" was deleted due to the detailed preparation of models were shown subsequently in 392-399.

  1. Comment: Page 15, line 274 - Why was 95 °C used to extract the volatile compounds from the tea? From the consumer's point of view, the recommended brewing temperature is between 70 and 90 °C.

Response: Thank you so much for the comment. From the consumer's point of view, the recommended brewing temperature for oolong tea, green tea and black tea is between 70 and 90 °C before drinking. However, the thing is different for white tea; it is usually cooked at boiling point in a teapot for a short period before drinking. From this point of view, the volatiles were extracted at 95 °C rather than 70-90 °C.

  1. Comment: Page 15, line 296 – What was the temperature during the evaporation of hexane to a volume of 0.5 mL?

Response: Thank you so much for your careful check. Because the solvent n-hexane is easy to vapor, a gentle stream of high-purity nitrogen was used to blow the extract so as to remove n-hexane at room temperature. In the revised manuscript, “The extract was concentrated approximately to the volume of 0.5 mL by using a gentle stream of high-purity nitrogen, and adjusted to the volume of 1.5 mL with hexane” was modified to “The extract was concentrated approximately to the volume of 0.5 mL at room temperature by using a gentle stream of high-purity nitrogen, and adjusted to the volume of 1.5 mL with hexane” (page 14, line 306).

  1. Comment: Page 15, line 297-298 - Is compared to loss of volatile compounds in the extract hexane to store the samples at -20 °C?

Response: Thank you so much for the comment. The purpose of storing the samples at -20 °C was not to protect the sample from vapor before analysis, rather than compare the loss of volatile compounds at -20 °C and room temperature. By following your instruction, the " The concentrated extraction was stored at -20 °C temporarily before analysis " was changed to " The concentrated extraction was stored at -20 °C temporarily before analysis with sensory evaluation, GC-MS and GC-O" in page 14, line 308-309.

All of the comments have been properly responded. Should you have any other questions or considerations, please feel free to contact us via [email protected].  Looking forward to hearing from you.

Yours sincerely,

Corresponding author: Professor Lijun Li

College of Food and Biological Engineering, Jimei University, Xiamen, Fujian Province 361021, China

Reviewer 2 Report

“Characterization of the aroma of an instant white tea dried by freeze drying”

By Hui Ni, Qing-Xiang Jiang, Ting Zhang, Gao-Ling Huang, Li-Jun Li and Feng Chen

The paper describes the aroma the aroma characterization of an instant white tea by steam distillation extraction coupled with GC-MS-O. Aroma active compounds were validated through aroma reconstruction and omission tests highlighting several synergic and antagonist components of the instant white tea.The manuscript fits the aims of the journal and of the topic of the special issue “Aromas and Volatiles of Food_2nd edition”. However, a revision of some parts are required.For this reason and details reported below, the manuscript deserve minor revision before publication. One of the main concern regards the samples analysed in the study, it is representative? In my opinion different samples, lots, batches should be testedAuthors very often in the text report the term “counterparts”, but it is not always clear to what counterparts their referred, please reviseTable 1 what it means IC? Why the authors do not give concentration in the same unit of the calibration curves? What it means ASC in the footnote of the table?Line 149 please use determination instead analysis, OAV is defined not analysedTable 2 NF? Not found?, Oder Characteristic should be OdorLines 198-200 are these sensory differences significant? Have the authors analysed panels data with anova?Paragraph 2.5 Figures should be cited ordered along the text, please revise the order and consequently the textIn conclusion counterparts of tea leaves infusion or of the spray drying WT?

Author Response

Dear Editors and Reviewers:

Thank you for your letter and for the reviewer’s comments concerning our manuscript entitled “Characterization of the aroma of an instant white tea dried by freeze drying” (ID: molecules-872960). We have made careful modifications according to the comments. The revisions are marked in red color. The modifications and responds to the comments are stated below one by one:

Responds to the comments of Reviewer #2

General comment: The paper describes the aroma the aroma characterization of an instant white tea by steam distillation extraction coupled with GC-MS-O. Aroma active compounds were validated through aroma reconstruction and omission tests, highlighting several synergic and antagonist components of the instant white tea. The manuscript fits the aims of the journal and of the topic of the special issue “Aromas and Volatiles of Food_2nd edition”. However, a revision of some parts are required. For this reason and details reported below, the manuscript deserve minor revision before publication.

Response: Thank you very much for the valuable comment. By following your comments and instrcutions, the manuscript was carefully modified.

  1. Comment: One of the main concern regards the samples analysed in the study, it is representative? In my opinion different samples, lots, batches should be tested.

Response: Thank you very much for raising the comment and pointing out the vague expression. It is a common sense that tea aroma varies with batches. To make representativeness, the instant white tea was made by mixing five different batches of fresh white teas. In the revised manuscript, the “A typical IWT was prepared with freeze drying from fresh white teas in Fujian Da Ming Development Company (Zhangzhou, Fujian province, China)” was changed to “A typical IWT was prepared with freeze drying from the mixture of five batch of fresh white teas in Fujian Da Ming Development Company (Zhangzhou, Fujian province, China)” (page 14, line 281).

  1. Comment: Authors very often in the text report the term “counterparts”, but it is not always clear to what counterparts their referred, please revise.

Response: Thank you very much for pointing out the vague expression. In the revision, the “counterparts” has been replaced the related chemicals as in page 2, line 64 and line 67; page 8, line 135 and line 168; page 17, line 410.

  1. Comment: Table 1 what it means IC? Why the authors do not give concentration in the same unit of the calibration curves? What it means ASC in the footnote of the table?

Response:  Thank you very much for pointing out the vague expression. CI is the abbreviation for "characteristic ion fragment", it has been revised in Table 1. The concentration units have been revised to mg/L throughout the manuscript. The revision in page 4, line 111-115; Table 1; page 7, line 130; Figure 2, line 212-214; page 16, line 379-380 and line 392-399 were marked in red. ASC has been noted at footnote “f” of Table 1 as the abbreviation for "authentic standard chemicals" (page 7, line 126).

  1. Comment: Line 149 please use determination instead analysis, OAV is defined not analysed.

Response:  Thank you very much for the instruction. In the revision, corrections have been made according to the comments, i.e., use determination instead analysis, and marked in page 8, line 153.

  1. Comment: Table 2 NF? Not found? Oder Characteristic should be OdorLines.

Response:  Thank you very much for pointing out the vague expression. NF means the threshold values have not been found in references, which has been notified in footnote “f” of the revised Table 2. Oder characteristic has been revised to aroma description in the revised Table 2.

  1. Comment: 198-200 are these sensory differences significant? Have the authors analysed panels data with anova?

Response:  Thank you very much for the instruction. The sensory values had been analyzed with ANOVA. In the revision, the p-values have been shown in the main text (page 11, line 187, 202, 204, 206) and the notes below Figure 1 (page 3, line 97-99) and Figure 2 (page 11, line 211-218).

  1. Comment: Paragraph 2.5 Figures should be cited ordered along the text, please revise the order and consequently the text.

Response:  Thank you very much for the instruction. In the revision, the results in Figure 3 were stated in the order of A, B, C and D. The details were shown in page 12, line 230-246 and line 255-264; page 13, line 273-277.

  1. Comment: In conclusion counterparts of tea leaves infusion or of the spray drying WT?

Response:  Thank you very much for pointing out the vague expression. In the revision, “The instant white tea and its SDE extract were dominated with floral and sweet notes, which is different from the counterpart of white teas.” was changed to “The freeze drying instant white tea and its SDE extract were dominated with floral and sweet notes, which is different from the aroma profiles of white tea leaves infusion” (page 17, line 409-410).

All of the comments have been properly responded. Should you have any other questions or considerations, please feel free to contact us via [email protected].  Looking forward to hearing from you.

Yours sincerely,

Corresponding author: Professor Lijun Li

College of Food and Biological Engineering, Jimei University, Xiamen, Fujian Province 361021, China

Round 2

Reviewer 1 Report

My suspension of the publication of this manuscript is due to the very high similarity to other works of these authors and the lack of scientific novelty in this research. It should be noted that the work is very similar to another article by the Authors, which also appeared in the Molecules journal in 2019 (Molecules. 2019 Dec; 24 (24): 4473; doi: 10.3390 / molecules24244473), in which the material for research was instant ripened Pu-erh tea. 

Author Response

Dear Editors and Reviewers:

Thank you for your concern on the novelty of the manuscript entitled “Characterization of the aroma of an instant white tea dried by freeze drying” (ID: molecules-872960), due to the work is very similar to another article by the Authors, which also appeared in the Molecules journal in 2019 (Molecules. 2019 Dec; 24 (24): 4473; doi: 10.3390 / molecules24244473). We carefully evaluated the novelty and similarity to our previous publications. By doing this, we are glad to conclude that the "molecules-872960" is almost nothing similar to "Molecules. 2019 Dec; 24 (24): 4473; doi: 10.3390 / molecules24244473", due to repeatability is less than 1%.

In particular, all the authors agree with the "molecules-872960" has three novelties. First, the experimental material (freeze-dried instant white tea) is different from, where the experimental material was instant ripened Pu-erh tea. Secondly, the volatile compounds detected in the two studies were different. The "Molecules. 2019 Dec; 24 (24): 4473; doi: 10.3390 / molecules24244473" showed linalool, linalool oxides, trans-β-ionone, benzeneacetaldehyde, and methoxybenzenes were the major aroma contributors to the SDE extract of instant ripened Pu-erh tea. While the "molecules-872960" showed the main aroma-active compounds in the SDE extract were camphene (green note, FD = 64), linalool (floral note, FD = 64), 2-hexanone (floral and sweet notes, FD = 16), trans-β-damascenone (sweet note, FD = 64), benzeneacetaldehyde (sweet and honey notes, FD = 16) and safranal (green and woody notes, FD = 16). The most interesting, the two studies showed there are different interactions of volatiles that played critical roles to determining the aroma of tea products. The "Molecules. 2019 Dec; 24 (24): 4473; doi: 10.3390 / molecules24244473" showed the stale note from methoxybenzenes had a reciprocal masking interaction with sweet, floral, and green notes, respectively. On the opposite, the "molecules-872960" showed linalool and trans-β-damascenone had a positive synergism interaction to promote both floral and sweet notes; and trans-β-damascenone and camphene had an antagonism interaction to reduce sweet and green notes.

Although the "molecules-872960" had some experimental methods similar to those in "Molecules. 2019 Dec; 24 (24): 4473; doi: 10.3390 / molecules24244473", it is nothing to lower the novelty of the new findings of "molecules-872960", because an effective experiment method always facilitates to solve various scientific problems.

From the three points of views, the "molecules-872960" had novelties that haven't been reported by previous studies of ourselves and other researchers.

Should you have any other questions or considerations, please feel free to contact us via [email protected].  Looking forward to hearing from you.

Yours sincerely,

Corresponding author: Professor Lijun Li

College of Food and Biological Engineering, Jimei University, Xiamen, Fujian Province 361021, China

Round 3

Reviewer 1 Report

Dear Authors;

Thank you for explaining what is new in the manuscript under review and for giving me detailed answers to my questions. I believe that, after supplementing and appropriate explanations, the article can be taken into account by Molecules.